# Knowledge, preventive behaviors and risk perception of the COVID-19 pandemic: A cross-sectional study in Turkish health care workers

Tufan Arslanca[1], Cihan Fidan[2], Mine Daggez[3], Polat Dursun[4]*

1 Department of Obstetrics and Gynecology, Ufuk University, Ankara, Turkey, 2 Department of Family Medicine, Başkent University, Ankara, Turkey, 3 Gynecologic Oncology Division, Department of Obstetrics and Gynecology, Erciyes University, Kayseri, Turkey, 4 Private Gynecologic Oncology Clinic, Ankara, Turkey

* pdursun@yahoo.com

**Data Availability Statement:** All relevant data are within the manucsript and its Supporting Information files.

**Funding:** The authors received no specific funding for this work.

## Abstract

The coronavirus disease 2019 (COVID-19) outbreak spread to over 100 countries with a total of 100,000 cases during the first week of March 2020. Health care workers, as those on the frontline of the COVID-19 pandemic, are more susceptible to infection. Inadequate related knowledge and preventive behaviors among health care workers might lead to delayed treatment and result in the rapid spread of the infection. Therefore, this study evaluated the knowledge of health care workers with regard to COVID-19. A cross-sectional study was conducted from June 10–18, 2020. Participants were general practitioners, specialists, and nurses working at the forefront of the pandemic. Their knowledge, preventive behaviors, and risk perceptions concerning COVID-19 were evaluated using an online questionnaire created by our medical specialists. The questionnaire consisted of 29, 5, and 4 items about COVID-19 knowledge, preventive behaviors, and risk perceptions, respectively. A total of 251 health care workers completed the questionnaire. The mean age of the participants was 33.88±8.72 years old, and the sample consisted of 68 males (27.08%) and 183 females (72.91%). While there was no difference between the percentage of correct answers given by female and male participants to knowledge-based questions (p>0.05), the percentage of correct answers to the questions on preventive behaviors was significantly higher in female participants than in males (p<0.001). The overall average percentages of correct responses were 91.66% for knowledge-based questions and 85.96% for preventive behavior questions. The scores for knowledge-based questions were higher for medical specialists, whereas nurses scored higher on preventive behavior questions. Government hospital staff showed a significant difference in preventive behaviors compared to that of university hospitals (p<0.05). In addition, there was a positive correlation between knowledge scores and preventive behaviors. Although all the participants (100%) knew that contracting COVID-19 can lead to death, only 66.93% of them were willing to get vaccinated themselves. The knowledge level of health care workers concerning COVID-19 was above 90%, but the level of competence in terms of preventive behaviors was found to be low, especially in males.

**Competing interests:** The authors have read the journal's policy and the authors have the following competing interests: PD does not have any commercial or institutional affiliation, but does operate a private gynecologic oncology practice. This does not alter our adherence to PLOS ONE policies on sharing data and materials. There are no patents, products in development or marketed products associated with this research to declare.

## Introduction

An outbreak of viral pneumonia of unknown etiology occurred in the city of Wuhan in eastern China in December 2019 [1]. This viral infection has received extensive attention throughout the country, as well as around the world [2]. The causative agent of the disease, named COVID-19 by the World Health Organization (WHO), is a coronavirus subtype called severe acute respiratory syndrome coronavirus 2 (SARS-CoV-2) [3]. The incubation period between the ingestion of the virus and the onset of symptoms is 2–14 days (4 days on average) [1, 4]. COVID-19 causes fatal pneumonia similar to that generated by severe acute respiratory syndrome coronavirus (SARS-CoV) and Middle East respiratory syndrome coronavirus (MERS-CoV), which have sporadically occurred in countries all over the world in the past 20 years [4]. Although COVID-19 is similar in many ways to SARS-CoV and MERS-CoV, it is also different in other ways. It may not be as severe as SARS-CoV and MERS-CoV. The crucial transmission route that is currently agreed upon is human-to-human via direct contact or respiratory droplets [5–7]. However, the rapid increase in incidence indicates that the virus is more contagious than SARS-CoV and MERS-CoV [4, 5, 7]. Also, there is no proven treatment or vaccination against SARS-CoV-2. Therefore, global concerns about the virus have risen dramatically [5].

According to the Ministry of Health of Turkey, the first detected case of COVID-19 in Turkey was announced on March 11, 2020. The first virus-related death in the country occurred on March 17, 2020. Confirmed cases are rising each day due to the increase in the number of laboratories that can perform a diagnostic or rapid test [8]. The pandemic poses a great risk for health care workers (HCWs), who are often in direct contact with infected patients [3]. While no HCWs in China had the disease at the beginning of the epidemic, it was reported that 7% of patients were HCWs after January 12, 2020 [9]. On April 1, 2020, Turkey reported that almost 600 infections of COVID-19 were of health personnel, while the rate for other patients was 3.6% [8]. The battle against COVID-19 is continuing in Turkey. To improve this situation, HCWs must have robust knowledge, attitudes, and practices pertaining to COVID-19.

Many international studies have found that the knowledge and attitudes of HCWs regarding infectious diseases are moderate [7, 10]. For example, only a rather low percentage of HCWs strictly follow the correct universal protection methods [11]. It is extremely important to protect HCWs from risk factors. For one, the infection of HCWs will adversely affect the supply of health care services, causing a decrease in the health care system's reaction to the epidemic and an uncontrolled increase in the incidence rate. Therefore, in this study, we determined the knowledge and protection levels of HCWs in relation to COVID-19 and evaluated their risk perceptions during the course of the pandemic.

## Materials and methods

This cross-sectional study was conducted in Turkey. The study's target population was HCWs working in pandemic hospitals in the center of Ankara. The 340 HCWs had established a common WhatsApp group to share current developments and news about the pandemic. Each participant had also shared their mobile phone number, email address, and further details to receive other information about the progress of COVID-19. We conducted our survey between June 10 and June 18, 2020, within this shared HCW WhatsApp group, which included doctors and nurses working at state or university hospitals, and 251 appropriate answers were received. The inclusion criteria were as follows: participants who were at least 18 years old, still working as HCWs (doctors and nurses), and with completed questionnaires. In line with measures to avoid the spread of COVID-19, the questionnaire was administered through the Internet.

## Demographics

This information included age, sex, level of education, marital status, current work status, and place of work. Participants were also asked if they had received any training on how to protect their wards from a COVID-19 outbreak. The demographic information section was designed based on a previous study [10].

## Knowledge of COVID-19

The scale for the level of knowledge consisted of a 29-item scale based on a previous study on MERS [10, 12]. In this section, there were three questions about COVID-19 etiology and basic science, eight about its transmission, eight about the incubation period and symptoms, four about treatment, and six about public prevention. For informational questions (vs. subjective questions), correct answers were given one point, and incorrect (and "I don't know") answers were given zero points.

## Preventive behaviors

The scale for evaluating preventive behaviors was based on previous studies [10, 13]. A five-item scale was used to assess responses; it including three items about preventive action during daily routines, one question about reducing the use of public spaces in daily life, and one question about conventional therapy methods for prevention.

The choices were "yes" or "no," and the participants were assigned one point for each positive behavior (wearing masks, using gloves, avoiding crowded environments, and using protective equipment in the workplace).

## Risk perception

The ultimate purpose of risk assessment is to limit an outbreak, enable emergency interventions, and mitigate the impact using non-drug public health measures. This is particularly important for COVID-19, which does not have currently a specific treatment or vaccine. Risk assessment starts with detecting the event and continues until it is under control [14]. Four questions were prepared to understand the risk assessment levels of HCWs. The questions asked whether they would get the COVID-19 vaccine, what they thought about the seriousness of COVID-19, whether they believed it will be contained, and finally, their opinions on how long it will take to contain it.

## Ethical considerations

This study was approved by the Dr. Abdurrahman Yurtaslan Ankara Oncology Educational and Research Hospital's Clinical Research Ethics Committee (2020–06/644) and the Ministry of Health Scientific Research Platform. Respondents' confidentiality and anonymity were ensured. The submission of the answered questionnaire was considered to be their consent to participate in the study.

## Sampling

Before starting the study, the reviewed literature and the minimum sample size determined by the publication's data were in accordance with the study's original hypothesis [15]. The sample size was calculated through power analysis using the G * Power 3.1.9.2 program based on gender groups in the publication [15]. With a 5% significance level (alpha), 95% confidence interval, and 80% power (1-beta), the required sample size was calculated as being at least 51 patients for each gender group.

### Statistical analysis

Descriptive data are given as average standard deviation, median, minimum, and maximum values, and continuous and discrete data are presented as percentages. The adherence of the data to a normal distribution was analyzed using the Kolmogorov–Smirnov test. The Mann–Whitney U test was used to compare the percentage of correct answers of participants with independent variables between two groups. The Kruskal–Wallis analysis of variance was used to compare independent variables for more than two groups. The groups originating from the differences were analyzed using the Kruskal–Wallis multiple comparison test. Chi-square and Fisher's exact tests were used for comparisons of nominal variables between groups. A general linear model test was used to compare the corrected data, while a multiple regression analysis was used to analyze the relationship between the corrected data. The data were analyzed using SPSS version 21.0 (IBM, Armonk, NY, USA). A p value of less than 0.05 was considered statistically significant.

## Results

A total of 251 HCWs responded to the questionnaire. The mean age of the participants was 33.88±8.72 years old (21–76). Overall, 72.91% of the participants were women, and 97.61% were university graduates. Of the health care professionals, 40.64% were nurses, 21.12% were general practitioners, and 38.24% were specialists. Of them, 40.24% were working in state hospitals and 59.76% in university hospitals. At the time of the survey, 68.92% of the health care professionals stated that they had received training about COVID-19.

The average percentage of correct answers to knowledge-based questions across all participants was 91.66±6.16 (65.5–100); it was 85.96±18.83 (25–100) for preventive behavior questions. The percentage of correct answers differed by occupation and place of work for knowledge-based questions; for preventive behavior questions, differences were found based on sex, education status, occupation, and workplace. There were no correlations between the age of participants and correct answers for either question group.

Medical specialists had a significantly higher percentage of correct answers to knowledge-based questions compared to general practitioners and nurses (p = 0.015 and p = 0.001, respectively), and public hospital employees significantly outperformed those working at university hospitals.

Regarding the preventive behavior questions, the percentage of correct answers was higher in females than in males, in high school graduates than in those with a university degree, in state hospital employees than in those working at university hospitals, and in nurses than in specialists. The rates of use of gloves, medical masks, N95 masks, visors/protective glass, disposable aprons/overalls, and aprons/jerseys were 95.62%, 92.82%, 35.85%, 20.72%, 12.35%, and 21.91%, respectively.

There were no differences in the rate of correct answers to knowledge-based and preventive behavior questions between those who had received COVID-19 training and those who had not (Table 1). Most participants (68.92%) stated that they had received training on COVID-19 at their institutions. This rate was higher at state hospitals (87.14%). However, more than half of the participants (53.44%) stated that they did not want to see patients with COVID-19 at their clinics. This rate was higher at university hospitals (58.71%).

Table 2 shows the responses of the participants to the knowledge-based, preventive behavior, and risk perception questions. There was no correlation between the age of participants and the percentage of correct answers to the questions on knowledge and attitude (p>0.05).

Except for Question 3 ("Do you believe that COVID-19 will eventually be contained?"), the risk perception differed according to the percentage of correct answers to the knowledge-

**Table 1. Responses according to sociodemographic and working characteristics.**

| | | Participants | Knowledge Score | Preventive Behaviors Score |
|---|---|---|---|---|
| | | n (%) | Mean±SD | Mean±SD |
| **Sex** | Female | 183 (72.91%) | 91.69±6.04 | 89.21±15.60 |
| | Male | 68 (27.09%) | 91.58±6.54 | 77.21±23.55 |
| | *P value* | | *0.917* | ***<0.001*** |
| **Education level** | High school | 6 (2.39%) | 85.63±11.83 | 100.0±0.00 |
| | University | 245 (97.61%) | 91.81±5.93 | 85.61±18.93 |
| | *P value* | | *0.189* | ***0.046*** |
| **Occupation** | Nurse[a] | 102 (40.64%) | 90.26±6.75 | 91.91±13.24 |
| | General practitioner[b] | 53 (21.12%) | 90.70±6.03 | 86.32±18.71 |
| | Medical specialist[c] | 96 (38.24%) | 93.67±4.99 | 79.43±21.76 |
| | *P value* | | ***0.001*** | ***<0.001*** |
| | *Posthoc test* | | ***a-c <0.01*** | ***a-c <0.001*** |
| | | | ***b-c <0.05*** | |
| **Workplace** | State Hospital | 101 (40.24%) | 93.10±5.52 | 88.86±18.87 |
| | University Hospital | 150 (59.76%) | 90.69±6.40 | 84.00±18.61 |
| | *P value* | | ***0.001*** | ***0.012*** |
| **Education on COVID-19** | Yes | 173 (68.92%) | 91.41±6.33 | 86.99±18.98 |
| | No | 78 (31.08%) | 92.22±5.79 | 83.65±18.39 |
| | *P value* | | *0.513* | *0.082* |

SD: standard deviation; statistically significant p values are in bold.

based and preventive behavior questions. For example, regarding Question 1 ("Would you like to get a COVID-19 vaccine?"), there were significant differences in the percentage of correct answers to the knowledge-based questions (hereafter, for clarity, CK%) among those who answered "yes," "no," and "I don't know," with it being significantly lower in those who answered "no" than in those who answered "yes" or "I don't know" (p = 0.000 and p = 0.033, respectively).

For Question 2 ("How do you judge the severity of COVID-19?"), there were no differences in the CK% between those who replied "very dangerous" or "moderately dangerous," but there were significant differences in the percentage of correct answers to the preventive behavior questions (CPB%) between these two groups, with it being significantly higher in the former group.

The results for Question 4 ("How long will it take before COVID-19 is contained?") were similar, with no differences in the CK% between those who answered "1–3 months," "3–6 months," or "6–12 months," but significant differences were observed in the CPB%, with it being significantly lower in the first group than in the other two groups (Table 3).

With regard to the knowledge scores of the dependent variable, the multiple linear regression model was found to be significant, in which the independent variables were age, sex, education, place of work, profession, and educational status ($R^2$ = 0.115, F = 4.494 p<0.001). When other variables are kept constant, working in the state hospital increases the knowledge scores by 2.013 points (p<0.05). In addition, if other variables are kept constant, the knowledge scores of those who are experts in their profession increase by 2.970 points compared to those of nurses (p<0.01). The knowledge scores of general practitioners were not found to be significant compared to those of the nurses (p>0.05) (Table 4).

In relation to the preventive behaviors scores of the dependent variable, the multiple linear regression model was found to be significant, in which the independent variables were age,

**Table 2. Correct answers to questions on knowledge, preventive behaviors, and risk perceptions.**

| Knowledge (True or False) | n | % |
|---|---|---|
| Have you heard of COVID-19? (T) | 251 | 100 |
| Does COVID-19 occur as a virus? (T) | 250 | 99.60 |
| Is COVID-19 transmitted by respiratory droplets? (T) | 250 | 99.60 |
| Can COVID-19 be transmitted while talking? (T) | 227 | 90.44 |
| Can COVID-19 be transmitted by shaking hands? (T) | 238 | 94.82 |
| Can COVID-19 be transmitted from animals to humans? (T) | 119 | 47.41 |
| Can diarrhea also appear in COVID-19 cases? (T) | 237 | 94.42 |
| Can COVID-19 lead to death? (T) | 251 | 100 |
| The incubation period for the virus is 4–6 days. (T) | 240 | 95.62 |
| Can COVID-19 patients recover completely? (T) | 228 | 90.84 |
| Can a COVID-19 patient have no complaints or symptoms? (T) | 232 | 92.43 |
| Does the COVID-19 contamination risk increase in crowded places? (T) | 250 | 99.60 |
| Can COVID-19 live on surfaces, such as door handles and tables, for a long time? (T) | 204 | 81.28 |
| Does hand washing protect against the virus? (T) | 251 | 100 |
| Does using a mask protect against the virus? (T) | 242 | 96.41 |
| The main symptoms of COVID-19 are a fever, a cough, shortness of breath, weakness, and muscle pain. (T) | 251 | 100 |
| A runny nose, nasal congestion, and sneezing are less frequent in COVID-19 infections. (T) | 201 | 80.08 |
| Currently, there is no effective treatment for COVID-19. (T) | 211 | 84.06 |
| Can a large proportion of COVID-19 infections be resolved without hospitalization? (T) | 250 | 99.60 |
| COVID-19 can be transmitted through close contact with or by eating wild animals. (T) | 163 | 64.94 |
| COVID-19 is transmitted from the infected patient via the respiratory tract. (T) | 237 | 94.42 |
| The purpose of the mask worn by non-hospital workers is to prevent them from getting infected with COVID-19. (T) | 174 | 69.32 |
| To avoid COVID-19, crowded areas and public transportation should not be used. (T) | 251 | 100 |
| Isolating patients with COVID-19 is effective for preventing the spread of the virus. (T) | 251 | 100 |
| The observation period of a patient with COVID-19 is 14 days. (T) | 241 | 96.02 |
| Older adults with chronic lung disease and obese people can have a more serious case of the disease. (T) | 250 | 99.60 |
| Other people will not be infected if a COVID-19-infected person does not have a fever. (F) | 234 | 93.23 |
| Young people and children do not need to take precautions to prevent infection. (F) | 245 | 97.61 |
| A COVID-19 vaccine exists. (F) | 242 | 96.41 |
| **Preventive Behaviors (Yes or No)** | **n** | **%** |
| Do you wear gloves? (Y) | 188 | 74.90 |
| Do you wear a mask when leaving the house? (Y) | 241 | 96.02 |
| Are you still going to crowded places? (Y) | 230 | 91.63 |
| Do you use alternative methods against COVID-19? (Y) | 204 | 81.28 |
| **Risk Perception** | **n** | **%** |
| 1. Would you like to get a COVID-19 vaccine? | | |
| Yes | 168 | 66.93 |
| No | 57 | 22.71 |
| I don't know | 26 | 10.36 |
| 2. How do you judge the severity of COVID-19? | | |
| Very dangerous | 174 | 69.32 |
| Moderately dangerous | 77 | 30.68 |
| 3. Do you believe COVID-19 will eventually be contained? | | |
| Yes | 211 | 84.06 |
| No | 13 | 5.18 |

*(Continued)*

**Table 2.** (Continued)

| | | |
|---|---|---|
| I don't know | 27 | 10.76 |
| 4. How long will it take before COVID-19 is contained? | | |
| 1–3 months | 44 | 17.53 |
| 3–6 months | 102 | 40.64 |
| 6–12 months | 105 | 41.83 |

sex, education, place of work, profession, and educational status ($R^2 = 0.159$, F = 6.587 $p<0.001$). As a result of multiple linear regression analysis, it was found that the explanatory factors for the preventive behavior scores were sex, place of work, and occupation variables. When other variables are kept constant, the female gender increases the preventive behavior scores by 8.407 points ($p<0.01$). When other variables are kept constant, the preventive behavior scores of those who are experts in their profession decrease by 9.957 points compared to those of nurses ($p<0.01$). The preventive behavior scores of general practitioners were not found to be significant compared to those of nurses ($p>0.05$) (Table 5).

## Discussion

Today, every country in the world is facing the COVID-19 pandemic. This pandemic has caused health problems, as well as high levels of anxiety and serious psychological problems for many. Undoubtedly, these effects are exacerbated in the health care field. Knowledge, attitudes, and practices pertaining to infectious diseases can affect the severity of the disease itself, the extent of spread, and the overall mortality rate [10, 12, 16]. It is extremely important to

**Table 3. Responses to the risk perception questions according to the percentage of correct answers to the knowledge-based and preventive behavior questions.**

| Risk Perception | Answer | Participants | Knowledge Score | Preventive Behavior Score |
|---|---|---|---|---|
| | | n (%) | Mean±SD | Mean±SD |
| Question 1 | Yes[a] | 168 (66.93%) | 92.63±5.49 | 85.71±18.87 |
| | No[b] | 57 (22.71%) | 89.11±6.00 | 83.33±20.23 |
| | I don't know[c] | 26 (10.36%) | 90.98±8.73 | 93.27±13.34 |
| | P value | | **0.001** | 0.087 |
| | Posthoc test | | **a-b <0.001** | |
| | | | **b-c <0.05** | |
| Question 2 | Very dangerous | 174 (69.32%) | 91.69±6.57 | 88.36±17.76 |
| | Moderately dangerous | 77 (30.68%) | 91.58±5.17 | 80.52±20.12 |
| | P value | | 0.233 | **0.002** |
| Question 3 | Yes[a] | 211 (84.06%) | 91.71±6.15 | 86.26±18.44 |
| | No[b] | 13 (5.18%) | 93.63±4.85 | 86.54±21.93 |
| | I don't know[c] | 27 (10.76%) | 90.29±6.69 | 83.33±20.80 |
| | P value | | 0.310 | 0.766 |
| Question 4 | 1–3 months[a] | 44 (17.53%) | 92.16±5.87 | 81.25±15.37 |
| | 3–6 months[b] | 102 (40.64%) | 91.65±5.56 | 86.03±20.98 |
| | 6–12 months[c] | 105 (41.83%) | 91.46±6.85 | 87.86±17.72 |
| | P value | | 0.769 | **0.023** |
| | Posthoc test | | | **a-b <0.05** |
| | | | | **a-c <0.05** |

SD: standard deviation; statistically significant p values are in bold.

**Table 4. Multiple regression analysis results of explanatory variables for knowledge scores.**

| | Knowledge Score | | | | | |
|---|---|---|---|---|---|---|
| Parameter | Estimate (β) | Std.Error | 95% CI | | t | p value |
| Constant | 84.717 | 3.112 | 78.586 | 90.847 | 27.220 | 0.000 |
| Age | -0.021 | 0.048 | -0.116 | 2.933 | -0.437 | 0.663 |
| Sex | 1.034 | 0.964 | -0.865 | 0.865 | 1.073 | 0.285 |
| Education level | 4.957 | 2.526 | -0.019 | 9.934 | 1.962 | 0.051 |
| Workplace | 2.013 | 0.856 | 0.327 | 3.698 | 2.932 | **0.019** |
| Occupation | | | | | | |
| General practitioner | 0.229 | 1.160 | -2.055 | 2.513 | 0.197 | 0.844 |
| Medical specialist | 2.970 | 1.013 | 0.974 | 4.965 | 2.932 | **0.004** |
| Education on COVID-19 | 0.840 | 0.931 | -0.993 | 2.674 | 0.903 | 0.368 |

Std: standard; CI: confidence interval

protect HCWs in order to control the outbreak. When these professionals become infected, it adversely affects the supply of health care services, causing a decrease in the efficiency of the health care system's reaction to the epidemic and an uncontrolled increase in the incidence rate. HCWs are on the frontline, and thus their knowledge and preventive behaviors are crucial to the success of any response. Increasing the knowledge of the disease is of great importance in reducing the level of anxiety of HCWs [17–19]. Thus, it is crucial to understand the current level of knowledge, preventive behaviors, and risk perceptions of HCWs.

This study addresses this knowledge gap regarding HCWs working in Turkey's capital city. We surveyed HCWs on the frontline of the pandemic: 40.64% of the participants were nurses, and the rest were doctors. Most of the doctors were specialists (64.43%). The medical specialists had the highest knowledge scores (p<0.001), followed by general practitioners and nurses. This highlights the importance of the role of medical doctors in combating the coronavirus.

The average score on knowledge-based questions was 91.66%, a rather high rate, whereas the average preventive behavior score was 85.96%. This is similar to in some studies [15, 17] but higher than those scores reported in other previous studies [12, 18, 20, 21]. In line with the literature, we found that the increase in the participants' knowledge level positively affects preventive behavior [10, 12, 18]. Overall, we found that roughly 69.32% of HCWs had received training on protecting themselves from the COVID-19 pandemic, such as through wearing

**Table 5. Multiple regression analysis results of explanatory variables for the preventive behavior scale.**

| | Preventive Behavior Score | | | | | |
|---|---|---|---|---|---|---|
| Parameter | Estimate (β) | Std.Error | 95% CI | | t | p value |
| Constant | 80.596 | 9.686 | 61.518 | 99.675 | 8.321 | 0.000 |
| Age | 0.241 | 0.143 | -0.041 | 0.523 | 1.686 | 0.093 |
| Sex | 8.407 | 2.868 | 2.757 | 14.057 | 2.931 | **0.004** |
| Education level | -12.126 | 7.517 | -26.933 | 2.680 | -1.613 | 0.108 |
| Workplace | 7.195 | 2.546 | 2.180 | 12.211 | 2.286 | **0.005** |
| Occupation | | | | | | |
| General practitioner | -0.736 | 3.450 | -7.533 | 6.060 | -0.213 | 0.831 |
| Medical specialist | -9.957 | 3.014 | -15.893 | -4.020 | -3.304 | **0.001** |
| Education on COVID-19 | 3.022 | 2.770 | -2.434 | 8.477 | 1.091 | 0.276 |

Std: standard; CI: confidence interval

preventive equipment during a patient triage at their institutions. This rate is low according to the literature [15, 17]. In our study, the main preventive behaviors outside of work were mask use (96% of HCWs) and the avoidance of crowded places (91.63%), while 74.90% used gloves. Regarding the use of protective materials during patient care, hospital gloves and medical masks were used at rates of 95.62% and 92.80%, respectively. Other protective materials (visors, protective glasses, and disposable aprons and uniforms, etc.) were used at a very low rate. Hence, it is crucial to implement strategies to increase professionals' knowledge of the disease and improve the protection of HCWs. Training is the primary strategy needed. HCWs need education and support when working on the frontline of pandemics.

Considering all the variables, it was determined that HCWs working in state hospitals and specialist doctors had better knowledge scores, and that women, HCWs working in state hospitals, and specialist doctors had better preventive behavior scores. Response rates did not differ by sex for the knowledge-based questions but did differ for the preventive behavior questions. In line with previous studies, females practiced more preventive behaviors [22, 23]. The participants' knowledge score average with a university or higher education level was significantly higher, but the expected high performance with regard to preventive behaviors was not observed. This situation suggests that the information is not reflected in terms of its application and that preventive behaviors should be audited regularly. One of the reasons for this may be the excessive workforce and a lack of adequate protective equipment support in our country.

In a previous study that measured knowledge and attitudes during the MERS outbreak, another disease belonging to the corona family, both scores increased with age [18, 22]. It was believed that this was due to both work and social experiences, which generally increase with age. In our study, no correlation was found between knowledge and preventive behavior and age. This is meaningful as we have not experienced a similar pandemic before.

The purpose of risk assessment is to limit the epidemic using various measures to control COVID-19, which does not currently have a specific treatment or vaccine. The risk perceptions of HCWs and society will be effective in controlling such infections. Most of the participants (69.32%) thought the COVID-19 infection to be hazardous, but the vast majority (84.06%) believed that this pandemic would be contained. Despite this, 58.17% of them did not think that it could be controlled within 6 months. In this study, the participants were not as optimistic as Chinese residents, maybe because pandemics are not common in our country [20]. This observation may be due to a lack of perceived information and knowledge relating to the COVID-19 pandemic. A recent study showed that both a high acceptance rate of a vaccine (90.60%) and a belief that one will eventually contract the disease (99.10%) demonstrate a definite inclination by individuals towards a vaccine as a prevention strategy [24]. However, only 66.93% of our participants were willing to get vaccinated against COVID-19. We may speculate that, although HCWs possess immense knowledge about the COVID-19 pandemic, they might have concerns about the adverse effects of the vaccination and other novel treatments.

One limitation of this study is that it only surveyed HCWs in a certain region, and therefore the results cannot be generalized. However, this study was carried out quickly to gain some early insights into perceptions and knowledge with regard to the pandemic. A study with a larger sample size will generate better results. Another limitation is that the questionnaire was prepared within a limited timeframe and when the pandemic was not very widespread in the country. It should also be kept in mind that the study results depend on participants' memories and honesty. So, this should be considered a preliminary study, and its results can be used to focus on effective risk communication and education on epidemic control.

In conclusion, we identified the knowledge and behavior statuses of HCWs who are at the frontline of the COVID-19 pandemic. According to our results, HCWs had a high level of

knowledge of COVID-19. Nevertheless, preventive behaviors against the global threat of this pandemic were not found to be at such a high level. It is necessary to monitor the implementation of preventive behaviors during the pandemic and provide the necessary, appropriate responses. More comprehensive studies should be conducted in the fight against the COVID-19 virus, which entails many obscurities. We suggest governments should publish guidelines based on practices modified according to the progression of the pandemic. Concurrently, improving risk perception and increasing preventive behaviors as well as commencing online sessions to help health professionals to evolve in terms of their understanding of the current guidelines and to protect themselves and others from being infected, will help diminish the spread of COVID-19.

## Supporting information

**S1 File. SPSS data of the survey.**
(SAV)

## Author Contributions

**Data curation:** Cihan Fidan, Mine Daggez, Polat Dursun.

**Formal analysis:** Tufan Arslanca, Cihan Fidan, Mine Daggez, Polat Dursun.

**Investigation:** Tufan Arslanca, Cihan Fidan, Mine Daggez.

**Methodology:** Tufan Arslanca, Mine Daggez, Polat Dursun.

**Resources:** Tufan Arslanca.

**Writing – original draft:** Tufan Arslanca, Cihan Fidan, Polat Dursun.

**Writing – review & editing:** Polat Dursun.

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
