## [Decision Letter · Decision Letter 0]

26 Aug 2020

PONE-D-20-23431

Knowledge, Attitudes and Practices of Turkish Healthcare Workers About SAR2-CoV-19 During Pandemic

PLOS ONE

Dear Dr. Dursun,

Thank you for submitting your manuscript to PLOS ONE. After careful consideration, we feel that it has merit but does not fully meet PLOS ONE’s publication criteria as it currently stands. Therefore, we invite you to submit a revised version of the manuscript that addresses the points raised during the review process.

We look forward to receiving your revised manuscript.

Kind regards,

Livia Melo Villar

Academic Editor

PLOS ONE

Additional Editor Comments:

Dear Author,

I have read your paper and the reviewer´s comments. Based on these comments, I suggested that you reviewed your manuscript and answered these comments to evaluate the new version of the paper.

Best

Livia

Journal Requirements:

We note that one or more of the authors are employed by a commercial company: Private Gynecologic Oncology.

3.1. Please provide an amended Funding Statement declaring this commercial affiliation, as well as a statement regarding the Role of Funders in your study. If the funding organization did not play a role in the study design, data collection and analysis, decision to publish, or preparation of the manuscript and only provided financial support in the form of authors' salaries and/or research materials, please review your statements relating to the author contributions, and ensure you have specifically and accurately indicated the role(s) that these authors had in your study. You can update author roles in the Author Contributions section of the online submission form.

3.2. Please also provide an updated Competing Interests Statement declaring this commercial affiliation along with any other relevant declarations relating to employment, consultancy, patents, products in development, or marketed products, etc.  

Please respond by return email with an updated Funding Statement and Competing Interests Statement and we will change the online submission form on your behalf.

Reviewers' comments:

Reviewer's Responses to Questions

**Comments to the Author**

1. Is the manuscript technically sound, and do the data support the conclusions?

Reviewer #1: Partly

Reviewer #2: Partly

Reviewer #3: Yes

Reviewer #4: Yes

Reviewer #5: No

2. Has the statistical analysis been performed appropriately and rigorously? 

Reviewer #1: Yes

Reviewer #2: No

Reviewer #3: Yes

Reviewer #4: Yes

Reviewer #5: No

3. Have the authors made all data underlying the findings in their manuscript fully available?

Reviewer #1: Yes

Reviewer #2: No

Reviewer #3: Yes

Reviewer #4: Yes

Reviewer #5: Yes

4. Is the manuscript presented in an intelligible fashion and written in standard English?

Reviewer #1: No

Reviewer #2: No

Reviewer #3: Yes

Reviewer #4: Yes

Reviewer #5: No

5. Review Comments to the Author

Reviewer #1: Review: The title of the article is not grammatically correct. Overall, substantive editing will be needed to address grammar, spelling, and syntax. The methods are not clear and require substantial improvements, the tables and order of reported information in the results might support a better understanding of the results, and both the introduction and discussion are confusing to read; primarily because of superfluous information that in no way relate to the topic or findings of the study.

Introduction: The context and scope are not clearly defined in the manuscript. The first two paragraphs and the introduction in general do not align well with the scope of the paper.

Methods: The methodology is unclear, was it a snowball sampling approach? Is Google Form used in other studies? Please report and cite how effective Google Form is and what protections there are to avoid double counting and ensure that people who are not HCWs are taking the survey. In demographics, can you clarify “the current status”? For the COVID-19 related knowledge questions, were these yes/no options or true/false questions. This is later defined in the results but should be adequately described in the methods section. The Cronbach’s alpha reported is quite low. This makes sense because you are measuring different domains. As you report the findings, it is unclear what these items actually measure. The composite score is not a single factor and therefore doesn’t really make sense to assess this way. Were there differences observed between domains?

Results: Table 1 should not include composite scores unless the instrument is better described in the methods. There is little information about what is included in the “knowledge score” or “preventative behaviors score”. Given that the details of these composite items are not outlined is difficult for me to understand exactly what the outcome is measuring (as it is written in Table 1). Table 2 and 3 are more informative and define the items – this is great, but comes too late in the manuscript to help follow the first portion of the results section. However, given that this is the first mention of the items it is confusing to read in the order it is placed. Consider updating the methods to better provide information about the items included in the analysis.

Discussion: Line 218 to 227 don’t have much value and contribute very little to this topic. There have been updated guidelines for mask use, please update the citation and information. The section that discusses use of face coverings and other masks don’t directly relate to the findings and seem out of place. Agreement to use one does not speak to efficacy and guidance of use. Paragraph between lines 270-279 are out of place and not relevant to the findings.

Reviewer #2: The authors have tried to assess the knowledge and preventive behaviors of health care workers (HCWs) in Turkey. However, the work requires major corrections and i will suggest the work be given o professional English language editors.

Kindly consider the following:

Title- Typographical error. SARS not SAR2. Consider revising the topic to : Knowledge and preventive behaviors of Turkish Health Workers to the COVID-19 Pandemic.

Introduction

Line 36- change "sporadic countries" to "sporadically in several countries"

Line 36 - Change are to were

Line 42- Since this is your first use of the Interquartile range (IQR), please define it first.

Line 44-45- Consider revision of the statement

Line 50- change first detected to index case

Line 53- In my opinion, a rapid test is still a type of diagnostic test, so please revise the statement

Line 53-54 - Consider removing the statement entirely "The battle against coVID-19..."

Line 54-56 - Consider revision of the paragraph "To guarantee...."

Line 57 - Change the word "leading force" to other synonyms like front-line workers/ personnel

Line 58- Include the word "COVID-19" before the word "related"

Line 59- Could you please explain how the lack of COVID-19 related knowledge leads to the overestimation of the situation, boost stress and anxiety and intermit medical decision.

Line 60- Use HCW(s) instead of healthcare workers as this has been introduced earlier in line 55.

Line 60- change exerted to implemented or instituted

Line 59-61 - consider revising the sentence "Precautions to be exerted..."

Line 62-63 - Consider revising the sentence "The COVID-19 contagion..." or remove it entirely.

Line 64- For consistency, please use HCWs instead of medical staff.

Line 63-65- Consider revising the sentence "therefore an essential study of medical staff knowledge..." or remove it entirely.

Line 66-69 - Consider rephrasing the aim of the study

Materials and Methods

I would suggest you include a sub-heading in line 71 (Study design- Just my suggestion)

Line 71- change the preposition "on" to "in'

Line 71 - change "their" to "the"

Line 72- Include the word "COVID-19" after relevant

Line 72- Change "this group" to "Participants in this study"

Line 73- Change includes to included (past tense)

Line 74- tautology of face and encounter (remove one and leave the other)

Line 73-75 - Consider revising the sentence "These workers are the first ones "

Line 76-78 - Consider revision of the statements. I would suggest " A structured questionnaire was administered via online social media platforms... "

Line 78- Remove 'The response rate' and take it to the results section.

Line 79 – Google forms are used to administer surveys not to design them

Line 86 – Include platforms after media and “remove to the contacts of…[[[[ ”

Line 156 – Include ‘graduates” after University

Line 161 – Rephrase the statement that started with “Conversely”

Line 180 - It is 78.35% not %78.35

Line 221-223 is repletion of results. Consider removing it entirely

Line 271 – Use A N95 not “An N95”

My concerns:

The population of HCWs in Turkey is 1,016,401 (Anadolu Agency, 2020). Do you think 305 respondents is representative of this population?

Could you kindly educate me on how you calculated the statistical power for the validity of a survey instrument and got 51 responses?

Did you calculate the odds ratio in the multivariable logistic regression?

How did you grade the responses? Who (or which group) had poor, average or satisfactory knowledge and behavior among your respondents?

There was no mention of practices in result (So, I suggest you remove practices from the topic).

What is the minimum threshold for reliability of variable in a survey instrument (using Cronbach’s alpha)?

Discussion:

Please revise the entire discussion.

Supplementary data: 6 files were indicated but only figure 1 was included on page 22.

Reviewer #3: I have reviewed the manuscript titled “Knowledge, Attitudes and Practices of Turkish Healthcare Workers About SAR2-CoV-19 During Pandemic” submitted to “PlosOne” for publication. In this paper, authors have conducted a survey study to investigate the level of COVID-19 related knowledge in health care workers in Turkey. This is a well conducted study and the manuscript is well presented and fits within the scope of the journal; it needs some major improvements; there are a few suggestions that authors may consider improving it further:

The use of English language is reasonable, however, there are a number of punctuation and grammatical errors; that should be corrected and rephrased using academic English for a better flow of text for reader. Authors may consider proofreading this manuscript.

Line 28: “In”?

I have no more comments regarding the introduction, methods and results as all sections have been well presented. The discussion section can be further improved. Surprisngly, authors missed inclusion of many recent studies that are similar to the subject. Here are a few examples;

COVID-19 Pandemic and Role of Human Saliva as a Testing Biofluid in Point-of-Care Technology. European Journal of Dentistry. 2020 Jun 3.

Knowledge and Attitude of Dental Practitioners Related to Disinfection during the COVID-19 Pandemic. Healthcare 2020, 8, 232.

Fear and practice modifications among dentists to combat Novel Coronavirus Disease (COVID-19) outbreak. International Journal of Environmental Research and Public Health. 2020 Jan;17(8):2821.

Author may discuss finding of such studies in context to improve the discussion.

There are a number of statements in the discussion require reference citations; please make sure, appropriate citations are in place.

What authors’ suggestions to further improve the level of COVID-19 related knowledge among HCW?

Line 308: educatşonal? Please correct

Reviewer #4: The authors evaluated personality variables, specific concerns, propose to self-isolate, and personal safety among healthcare workers practicing in Turkey during the current COVID-19 pandemic. The study is interesting and needs minor revision.

In the text, change outbreak to pandemic

The authors wrote “determined mid-level competence in terms of preventive behavior”. What is consider a mind-level?

Explain in the text which is convenient personal protective equipment (PPE)

The description of title of all tables needs to be improved to reflect the data analyzed in the tables.

The layout of all tables needs to be reformulated, since the sides of tables should be open, without internal lines.

It is not clear if when the question “training on how to protect you from the COVID-19” was donned, if it was explained which type of training the authors had referred.

Discussion

The authors suggested “During the pandemic, it is crucial to implement strategies that will increase knowledge and improve the protection of HCWs, and training is the main point of these strategies”. Write in the manuscript the strategies that are crucial

Conversely, only 66.2% of our participants willing to have themselves COVID-19 vaccine implemented. This answer should be more discussed and compared to data available in the period that this study was conducted.

All results that were signifcative in the multiple linear regression shoud be disccused.

In the abstract the authors wrote: the most commonly defined sources of knowledge were Television (69.8%), Twitter (34%), followed by Instagram (24.5%), Facebook (23.6%). However, this data was not discussed. The authors should discussed the influence of social medias and TV as objects of information in the context of COVID-19.

Finally, the authors must make one more update in the current literature, considering the number of articles that are published on the topic daily.

Reviewer #5: Study is flawed by the methodology, science behind the study, analysis and discussion was largely a repeat of the introduction. I have provided detailed guideline on what authors can do to improve the study if they wish to

6. PLOS authors have the option to publish the peer review history of their article (what does this mean?). If published, this will include your full peer review and any attached files.

Reviewer #1: No

Reviewer #2: No

Reviewer #3: No

Reviewer #4: No

Reviewer #5: No

---

## [Author Response · Author response to Decision Letter 0]

4 Nov 2020

Thank you for the editors' comments. We have altered the whole manuscript contingent upon the statistics changes. We hope that this manuscript will lead the healthcare workers with a novel perspective.

---

## [Decision Letter · Decision Letter 1]

29 Dec 2020

PONE-D-20-23431R1

Knowledge, preventive behaviors and risk perception of the COVID-19 pandemic: A cross-sectional study in Turkish health care workers

PLOS ONE

Dear Dr. Dursun,

Thank you for submitting your manuscript to PLOS ONE. After careful consideration, we feel that it has merit but does not fully meet PLOS ONE’s publication criteria as it currently stands. Therefore, we invite you to submit a revised version of the manuscript that addresses the points raised during the review process.

We look forward to receiving your revised manuscript.

Kind regards,

Livia Melo Villar

Academic Editor

PLOS ONE

Additional Editor Comments (if provided):

Dear Author,

I recommend major revision of this paper as suggested by the reviewers,

Sincerely,

Reviewers' comments:

Reviewer's Responses to Questions

**Comments to the Author**

1. If the authors have adequately addressed your comments raised in a previous round of review and you feel that this manuscript is now acceptable for publication, you may indicate that here to bypass the “Comments to the Author” section, enter your conflict of interest statement in the “Confidential to Editor” section, and submit your "Accept" recommendation.

Reviewer #1: All comments have been addressed

Reviewer #2: (No Response)

Reviewer #3: (No Response)

Reviewer #5: (No Response)

2. Is the manuscript technically sound, and do the data support the conclusions?

Reviewer #1: Partly

Reviewer #2: Partly

Reviewer #3: Yes

Reviewer #5: No

3. Has the statistical analysis been performed appropriately and rigorously? 

Reviewer #1: N/A

Reviewer #2: I Don't Know

Reviewer #3: No

Reviewer #5: No

4. Have the authors made all data underlying the findings in their manuscript fully available?

Reviewer #1: Yes

Reviewer #2: Yes

Reviewer #3: Yes

Reviewer #5: Yes

5. Is the manuscript presented in an intelligible fashion and written in standard English?

Reviewer #1: Yes

Reviewer #2: No

Reviewer #3: Yes

Reviewer #5: No

6. Review Comments to the Author

Reviewer #1: The resubmission of this manuscript has addressed several comments and also improved the language in the text. However, there are still a few typos in the document as well as concerns that the logical translation may not accurately describe the authors intent. There appears to also be a lack of consistency (e.g., on how many decimals are used for percentages) throughout the manuscript. Generally, it reads as though different people wrote different sections. One example of this is that there is no mention that a KAP survey is being implemented in the methods section, though the discussion makes strong mention of a KAP approach.

Reviewer #2: (No Response)

Reviewer #3: I am afraid, my comments are not responded; may be authors did not get the comments previously. Therefore, I am pasting the same comments again and would like to request the editor to give another oppurtunity to authors to respond to comments; many thanks:

Comment:

I have reviewed the manuscript titled “Knowledge, Attitudes and Practices of Turkish Healthcare Workers About SAR2-CoV-19 During Pandemic” submitted to “PlosOne” for publication. In this paper, authors have conducted a survey study to investigate the level of COVID-19 related knowledge in health care workers in Turkey. This is a well conducted study and the manuscript is well presented and fits within the scope of the journal; it needs some major improvements; there are a few suggestions that authors may consider improving it further:

The use of English language is reasonable, however, there are a number of punctuation and grammatical errors; that should be corrected and rephrased using academic English for a better flow of text for reader. Authors may consider proofreading this manuscript.

Line 28: “In”?

I have no more comments regarding the introduction, methods and results as all sections have been well presented. The discussion section can be further improved. Surprisngly, authors missed inclusion of many recent studies that are similar to the subject. Here are a few examples;

COVID-19 Pandemic and Role of Human Saliva as a Testing Biofluid in Point-of-Care Technology. European Journal of Dentistry. 2020 Jun 3.

Knowledge and Attitude of Dental Practitioners Related to Disinfection during the COVID-19 Pandemic. Healthcare 2020, 8, 232.

Fear and practice modifications among dentists to combat Novel Coronavirus Disease (COVID-19) outbreak. International Journal of Environmental Research and Public Health. 2020 Jan;17(8):2821.

Author may discuss finding of such studies in context to improve the discussion.

There are a number of statements in the discussion require reference citations; please make sure, appropriate citations are in place.

What authors’ suggestions to further improve the level of COVID-19 related knowledge among HCW?

Line 308: educatşonal? Please correct

Reviewer #5: There are methodological issues with the current manuscript and some reports of 100% survey returned despite using online distribution platform, lack of incentive and participation was voluntary and no professional body helped with providing emails of members is unusual.

7. PLOS authors have the option to publish the peer review history of their article (what does this mean?). If published, this will include your full peer review and any attached files.

Reviewer #1: No

Reviewer #2: No

Reviewer #3: No

Reviewer #5: No

---

## [Decision Letter · Decision Letter 2]

30 Mar 2021

Knowledge, preventive behaviors and risk perception of the COVID-19 pandemic: A cross-sectional study in Turkish health care workers

PONE-D-20-23431R2

Dear Dr. Dursun,

We’re pleased to inform you that your manuscript has been judged scientifically suitable for publication and will be formally accepted for publication once it meets all outstanding technical requirements.

Kind regards,

Livia Melo Villar

Academic Editor

PLOS ONE

Additional Editor Comments (optional):

Dear Author,

After reading the revised version and comments of the reviewers, I recommend the acceptance of this paper,

Best regards,

Livia

Reviewers' comments:

Reviewer's Responses to Questions

**Comments to the Author**

1. If the authors have adequately addressed your comments raised in a previous round of review and you feel that this manuscript is now acceptable for publication, you may indicate that here to bypass the “Comments to the Author” section, enter your conflict of interest statement in the “Confidential to Editor” section, and submit your "Accept" recommendation.

Reviewer #3: All comments have been addressed

2. Is the manuscript technically sound, and do the data support the conclusions?

Reviewer #3: Yes

3. Has the statistical analysis been performed appropriately and rigorously? 

Reviewer #3: Yes

4. Have the authors made all data underlying the findings in their manuscript fully available?

Reviewer #3: Yes

5. Is the manuscript presented in an intelligible fashion and written in standard English?

Reviewer #3: Yes

6. Review Comments to the Author

Reviewer #3: Dear Authors, Many thanks for the revision and incorporating all suggested changes to the manuscript

7. PLOS authors have the option to publish the peer review history of their article (what does this mean?). If published, this will include your full peer review and any attached files.

Reviewer #3: No

---

## [Editor Report · Acceptance letter]

1 Apr 2021

PONE-D-20-23431R2 

Knowledge, preventive behaviors and risk perception of the COVID-19 pandemic: A cross-sectional study in Turkish health care workers 

Dear Dr. Dursun:

I'm pleased to inform you that your manuscript has been deemed suitable for publication in PLOS ONE. Congratulations! Your manuscript is now with our production department. 

Kind regards, 

on behalf of

Dr. Livia Melo Villar 

Academic Editor

PLOS ONE